# Cobalt Iron Oxide (CoFe_2_O_4_) Nanoparticles Induced Toxicity in Rabbits

**DOI:** 10.3390/vetsci10080514

**Published:** 2023-08-09

**Authors:** Muhammad Shahid Khan, Saeed Ahmad Buzdar, Riaz Hussain, Abdulaziz Alouffi, Muhammad Tahir Aleem, Muhammad Farhab, Muhammad Arshad Javid, Rana Waseem Akhtar, Iahtasham Khan, Mashal M. Almutairi

**Affiliations:** 1Institute of Physics, The Islamia University, Bahawalpur 63100, Pakistan; mshahid.khan@iub.edu.pk (M.S.K.); saeed.buzdar@iub.edu.pk (S.A.B.); arshad.javid@iub.edu.pk (M.A.J.); 2Department of Pathology, Faculty of Veterinary and Animal Sciences, The Islamia University, Bahawalpur 63100, Pakistan; 3King Abdulaziz City for Science and Technology, Riyadh 12354, Saudi Arabia; asn1950r@gmail.com; 4Center for Gene Regulation in Health and Disease, Department of Biological, Geological and Environmental Sciences, College of Sciences and Health Professions, Cleveland State University, Cleveland, OH 44115, USA; dr.tahir1990@gmail.com; 5MOE Joint International Research Laboratory of Animal Health and Food Safety, College of Veterinary Medicine, Nanjing Agricultural University, Nanjing 210095, China; 6Key Laboratory of Animal Genetic Engineering, College of Veterinary Medicine, Yangzhou University, Yangzhou 225009, China; farhab.dvm@gmail.com; 7Jiangsu Co-Innovation Center of Prevention and Control of Important Animal Infectious Diseases and Zoonoses, Yangzhou University, Yangzhou 225009, China; 8Department of Animal Breeding and Genetics, Faculty of Veterinary and Animal Sciences, The Islamia University, Bahawalpur 63100, Pakistan; dr.ranawasim2@gmail.com; 9Section of Epidemiology and Public Health, Department of Clinical Sciences, College of Veterinary and Animal Sciences, Jhang Sub-Campus University of Veterinary and Animal Sciences, Lahore 54000, Pakistan; iahtasham.khan@uvas.edu.pk; 10Department of Pharmacology and Toxicology, College of Pharmacy, King Saud University, Riyadh 11451, Saudi Arabia

**Keywords:** cobalt iron oxide (CoFe_2_O_4_) nanoparticles (CIONPs), XRD, rabbits, magnetic resonance imaging, relaxivity, blood biochemistry, histopathology

## Abstract

**Simple Summary:**

This study was carried out to ascertain the toxicity of synthetic cobalt iron oxide (CoFe_2_O_4_) nanoparticles (CIONPs) in rabbits. Sixteen rabbits in total were purchased from the neighborhood market and divided into two groups (A and B), each of which contained eight rabbits. The CIONPs were synthesized by the co-precipitation method and were administrated intravenously into the rabbits through the ear vein. Blood was collected at days 5 and 10 post-exposure for hematological and serum biochemistry analyses. Different histological ailments were also observed in the visceral organs of treated rabbits. Cobalt iron oxide (CoFe_2_O_4_) nanoparticles appeared to induce toxicity in rabbits.

**Abstract:**

The market for nanoparticles has grown significantly over the past few decades due to a number of unique qualities, including antibacterial capabilities. It is still unclear how nanoparticle toxicity works. In order to ascertain the toxicity of synthetic cobalt iron oxide (CoFe_2_O_4_) nanoparticles (CIONPs) in rabbits, this study was carried out. Sixteen rabbits in total were purchased from the neighborhood market and divided into two groups (A and B), each of which contained eight rabbits. The CIONPs were synthesized by the co-precipitation method. Crystallinity and phase identification were confirmed by X-ray diffraction (XRD). The average size of the nanoparticles (13.2 nm) was calculated by Scherrer formula (Dhkl = 0.9 λ/β cos θ) and confirmed by TEM images. The saturation magnetization, 50.1 emug^−1^, was measured by vibrating sample magnetometer (VSM). CIONPs were investigated as contrast agents (CA) for magnetic resonance images (MRI). The relaxivity (r = 1/T) of the MRI was also investigated at a field strength of 0.35 T (Tesla), and the ratio r_2_/r_1_ for the CIONPs contrast agent was 6.63. The CIONPs were administrated intravenously into the rabbits through the ear vein. Blood was collected at days 5 and 10 post-exposure for hematological and serum biochemistry analyses. The intensities of the signal experienced by CA with CIONPs were 1427 for the liver and 1702 for the spleen. The treated group showed significantly lower hematological parameters, but significantly higher total white blood cell counts and neutrophils. The results of the serum biochemistry analyses showed significantly higher and lower quantities of different serum biochemical parameters in the treated rabbits at day 10 of the trial. At the microscopic level, different histological ailments were observed in the visceral organs of treated rabbits, including the liver, kidneys, spleen, heart, and brain. In conclusion, the results revealed that cobalt iron oxide (CoFe_2_O_4_) nanoparticles induced toxicity via alterations in multiple tissues of rabbits.

## 1. Introduction

Cobalt iron oxide is the most extensively researched ferrite after Fe_3_O_4_. It is a hard magnetic material with a mixed or inverse spinel structure. Its characteristics, like those of other ferrites, vary with the synthesis process, impurities, size, and temperature [1,2,3]. Due to their unique properties, CIONPs and various other nanoparticles are being used in numerous applications comprising tissue imaging, magnetic resonance imaging (MRI), drug delivery, cancer therapy, improving meat quality, antioxidants, and antimicrobial and antiparasitic agents [4,5,6]. Cobalt ferrite has the strongest spin-orbital coupling and anisotropy constant when compared to other MNPs. These features, together with the others stated above, make CoFe_2_O_4_ one of the most significant NPs for biological applications [7,8,9]. In the biological system, nanoparticles have been widely studied as therapeutic agents and carriers for medical applications [10]. CIONPs have become one of the most widely investigated magnetic material due to their excellent heat-generating potential and exceptional mechanical and chemical stability, as well as being cost-effective [11,12]. Magnetic particles have been found to be favorably responsive towards lungs cancer cells [13], red blood cells [14,15], and urological cancer cells [16,17]. These MNPs are valuable in the field of life sciences, such as bio-imaging, bio-sensing, and drug administration systems, due to their good magnetic properties, low toxicity, and small diameter [18,19]. The use of hyperthermia in cancer therapy has expanded significantly. These MNPs used in biological applications should be super paramagnetic at room temperature [20,21,22]. CIONPs have received much interest and attention from many researchers because of the slow magnetic moment relaxation of CoFe_2_O_4_ compared to Fe_3_O_4_ nanoparticles. As a result, prior to their biomedical applications, it is perilous to decipher the biological interactions of CIONPs at the molecular level.

At the MRI level, the relaxation durations, T_1_ (spin–lattice relaxation, longitudinal) and T_2_ (spin–spin relaxation, transverse), are both effected by the quantity of protons in water molecules [19,23] that interact magnetically and the nanoparticles’ Ms value. To minimize the relaxation time parameter following tissue differentiation, various relocation speed limitations for tissues and contrast agents are applied [23,24]. At a normal temperature, cobalt nanoparticles have a substantially greater Ms value than iron oxides, which might result in a stronger influence on proton relaxation. This increases MRI contrast and enables the use of smaller magnetic nanoparticle cores while maintaining sensitivity. The r_1_ relaxivity of larger magnetic particles is greater than that of smaller magnetic particles. The field intensity and particle size have no influence on r_2_ relaxivity, and because r_2_ is rather high, Co nanoparticles qualify as a negative contrast agent.

There have been only a few studies reporting the toxic effect of CIONPs. Various kinds of cells showed diverse dose–response curves for the cell viability after the exposure of CIONPs. Majeed and Ou et al. [25,26] observed in guinea pigs that CIONPs induce respiratory problems. Hadrup and Ansari et al. [27,28] reported that CIONPs cause micronuclei to develop in human peripheral lymphocytes. EI-Hamaky et al. [29] reported that CIONPs have also been discovered to interfere with lipid metabolism and embryogenesis. However, Ahmad et al. [30] reported that CIONPs were not hazardous to human pancreatic and ovarian cancer cells and might be a useful tool for hyperthermia and tumor identification. Furthermore, CIONPs were found to be hazardous to zebrafish and green algae [31]. According to previous studies, MNPs have been proven to cause toxicity in human cells by causing oxidative damages to cell macromolecules [19,32,33,34].

Various earlier reports on the monitoring of different nanomaterials such as nickel iron oxide, copper iron oxide, zinc iron oxide, calcium, and copper NPs have indicated cytotoxic effects in different species, including human cells, via oxidative stress [19,35,36,37,38,39,40,41]. The CIONPs used in our study are an example of non-surface functionalized behaviour, which occurs when a surface is not modified in any way to acquire physical, chemical, or biological properties different from those it already possessed. Such materials do not have conjugation or reactive sites that facilitate the formation of covalent bonds with other molecules. Therefore, we used these nanoparticles to know the toxic effects in rabbits because the physiological functions of these animals closely resemble to human beings. Little information could be found regarding the underlying mechanisms of toxicity of CIONPs. Therefore, this study investigated the biocompatibility of CIONPs and their adverse effects on different organs, such as the kidneys, liver, spleen, heart, and brain of rabbits (*Oryctolagus cuniculus*). The CIONPs were injected into the rabbits following the previous experimental protocol [19,26] for hematological, serum biochemical, and histopathology of selected organs of the controlled and treated rabbits.

## 2. Materials and Methods

### 2.1. Chemicals and Preparation of Nanomaterials

All the commercially available chemicals used in this study were obtained from Sigma-Aldrich and were utilized without any further purification. The co-precipitation technique was used to prepare the nanomaterials. Briefly, ferrous chloride tetrahydrate (FeCl_2_-4H_2_O), ferric chloride hexahydrate (FeCl_3_-6H_2_O), and cobalt chloride hexahydrate (CoCl_2_-6H_2_O) were combined in a 2:1 ratio in a beaker and were placed on a magnetic stirrer plate at 40 °C and 500 rpm for 10 min. The pH (9.0) of the solution was sustained by adding ammonium hydroxide (NH_4_OH). After that, the solution was centrifuged for 15 min at 5000 rpm to obtain the nanoparticles. The nanoparticles were then dried for 30 min at 60 °C in an oven. All physical and chemical operating requirements were met and the CIONPs were successfully obtained [42].

### 2.2. X-ray Diffraction (XRD) Analysis

X-ray diffraction (XRD) analysis confirmed the crystalline structure of the prepared nanoparticles. The crystallinity, average particle size, and phase identification of nanoparticles were studied using a Bruker-D8 advance laboratory diffractometer (Bruker Corporation, Massachusetts, United Stated of America), with CuKα1 radiation, λ = 1.54 Å). The Scherrer equation (Dhkl = 0.9 λ/β cos θ) was used to examine the average crystallite size of the samples [43].

### 2.3. Transmssion Electron Microscope (TEM) Analysis

Transmission electron microscopy (TEM) analysis was used to investigate the morphology of the CIONPs. The size of the CIONPs was also confirmed. To estimate their mean particle size, a statistic histogram plot was created using a normal fit tool [44].

### 2.4. Vibrating Sample Magnetometer (VSM) Analysis

A vibrating sample magnetometer (VSM) confirmed the magnetic behavior of the synthesized material. The saturation magnetization of the hysteresis loop was investigated using VSM Lakeshore 7407 (Lakeshore cryotronics, Westerville, OH, USA) [45].

### 2.5. Magnetic Resonance Imaging (MRI) Contrast Agent (CA) and Relaxivity Analysis

Magnetic resonance imaging (MRI) was used to observe the contrast effect of CIONPs in the kidneys and liver of rabbits. The protocols for MRI were followed as per previous literature [36,43]. MRI was used to investigate the intensity of a region of interest in the organs of rabbits. As described in previous studies [42], the relaxivity of the spleen and liver was also determined using MRI data.

### 2.6. Experimental Animals

Sixteen rabbits (*n* = 16; *Oryctolagus cuniculus*) of the same weight and age were obtained from a local private market in the district of Bahawalpur, Pakistan. The animal study was reviewed and approved by Institutional Animal Ethics Procedures and Guidelines of the Islamia University of Bahawalpur (IUB), Pakistan (Protocol No. 759). They were kept in similar conditions in the laboratory, having unlimited access to fresh water and food. The house was cleaned and disinfected prior to the start of the experiment. The rabbits were randomly separated into two groups, an untreated control group (A) and a treatment group (B), each having 8 rabbits after 5 days of acclimatization. CIONPs were given to rabbits of group B for a period of 10 days. MRI contrast agent evaluation was performed using the rabbits. Later on, the rabbits were anesthetized (ketamine salts) to restrain them for further procedures accordingly [46,47]. The CIONPs solution was prepared in a 10 mL saline solution. To see the contrast in the liver and spleen, a 0.1 mg/kg dose of contrast agent was injected into each rabbit through an ear vein. The rabbits’ MRI was performed at a low field of 0.35 T (magnetron). The contrast enhancement on the MRI images was compared using four rabbits of the same weight [19]. Similarly, the contrast agent’s relaxivity was evaluated. The rabbits were taken for further study for histopathology, hematological, and serum biochemistry after 10 days of treatment. Hematological and serum biochemical tests were performed by taking approximately 2.5 mL blood collected on days 5 and 10 post-therapy. At days 5 and 10 after exposure, 4 rabbits from each group were slaughtered and blood samples were taken from the jugular vein in test tubes containing an anticoagulant, whereas for serum analysis, all blood samples from each rabbit were taken in glass test tubes without an anticoagulant. Various blood indicators were measured, as previously reported [48]. The serum biochemical parameters, including alanine transaminase (ALT), lactate dehydrogenase (LDH), alkaline phosphatase), aspartate transaminase (AST), creatinine, urea, cholesterol, triglycerides, and glucose, were determined according to prior protocols using commercially available kits [48,49].

### 2.7. Gross and Histopathology

After blood collection, the rabbits were euthanized and various visceral organs, such as the spleen, brain, kidneys, heart, and liver, were carefully observed for any obvious lesions and removed. A small piece of each visceral tissue was preserved in a 10% formaldehyde solution for fixation and further histological studies. Finally, all the collected tissues were processed using standard histopathological procedures and were stained with hematoxylin and eosin [48,50].

### 2.8. Statistical Analysis

The data obtained on the blood profile and serum biochemistry in this trial were analyzed using an analysis of variance (ANOVA) and *t*-test utilizing using IBM SPSS. *p* ≤ 0.05 was considered a significant level.

## 3. Results

### 3.1. XRD, Transmission Electron Microscope (TEM), and VSM Analysis

Figure 1 depicts the XRD and TEM image, whereas Figure 2 presents the histogram and VSM plot results of synthesized CoFe_2_O_4_. The acquired characteristics peaks matched perfectly to the standard pattern of cubic CoFe_2_O_4_ (JCDPS card no. 22-1086). The corresponding hkl values for various peaks at 2 theta 18.20, 30.00, 35.30, 43.10, 53.40, 56.80, 62.50, 70.90, 74.00, and 78.90 are (111), (220), (311), (222), (400), (422), (511), (440), (531), and (622), respectively. The average size of the nanoparticles (13.2 nm) was calculated by the Scherrer formula (Dhkl = 0.9 λ/β cos θ). TEM images showed the structural morphology of the synthesized materials. Black layers on the cubic-shape nanoparticles showed the coating of cobalt onto the iron nanoparticles. Since the TEM image contains several nanoparticles of varying sizes, approximately 25 particles were selected in a single TEM image to determine the average size of the particles. From the TEM image, the particle size (13.4 nm) was also confirmed. According to the previous literature, room temperature was used. These MNPs employed in biomedical applications had a super paramagnetic nature [51]. The magnetic behavior of the synthesized nanoparticles was confirmed by VSM. The results indicated that CIONPs have an excellent saturation magnetization value of about 50.1 emug^−1^ and the coercivity is almost zero. The magnetization hysteresis reveals that the CIONPs had a super paramagnetic nature (Figure 1 and Figure 2).

### 3.2. MRI Contrast Agent and Relaxivity

CIONPs are presumed to be a promising candidate for the development of a T_2_ contrast agent with better relaxivity. CIONPs essentially disrupted the magnetic relaxation process of protons in the tissue, causing the proton’s spin–spin relaxation time to shorten. The relatively high magnetization caused the relaxivity of the body’s underwent tissue to improve. In this investigation, MRI was performed at a low-field MRI magnetron (0.35 T) in order to obtain high contrast in the organs, such as the rabbit’s liver and spleen. An intravenous dose of the contrast agent was administered in the ear vein. We evaluated the intensity of the signal experienced by the contrast agent as with CIONPs, using IQ View software (for liver, I = 1427 S.D = 121.1, spleen I = 1702 S.D = 221.3), as shown in Figure 3 and Figure 4. The slope of the line fitted to the 1/T_2_ versus concentration plot was used to calculate proton relaxivity (r_2_) (d). CIONPs had a r_2_ value of 112.4 mM^−1^ S^−1^. The 1/T_1_ value was shown as a function of the varied concentrations plot (Figure 4C) to confirm that CIONPs exhibited a negative T_2_-type of contrast. When compared to r_2_, the r_1_ relaxivity is minimal.

### 3.3. Hematological and Serum Analysis

The results on various blood profiles, including red blood cell count, white blood cell counts, hemoglobin quantity, hematocrit value, neutrophil counts, monocyte, and lymphocyte counts, are shown in Figure 5. Comparison of hematological profile of normal and cobalt iron oxide (CoFe_2_O_4_) nanoparticles (CIONPs) treated rabbits can be found in the Appendix A.

The hematological results showed that the erythrocyte counts, hematocrit percentage, and hemoglobin concentrations were significantly reduced in CIONPs-exposed rabbits at day 10 of the experiment. The results showed significantly higher total white blood cell counts and a higher neutrophil population compared to control rabbits at different intervals of the experiment. The values of monocyte and lymphocyte counts also significantly reduced in treated rabbits compared to the control. The results on different serum biochemical analyses and antioxidant enzymes and oxidative stress biomarkers are presented in Figure 6 and Figure 7. Serum biochemistry profile, and Antioxidant enzymes and oxidative stress biomarkers of normal and cobalt iron oxide (CoFe_2_O_4_) nanoparticles (CIONPs) treated rabbits can be found in the Appendix A. The quantity of serum alanine aminotransferase, alkaline phosphatase, urea, creatinine, triglyceride, cholesterol, and glucose increased significantly on day 10 in treated rabbits compared to the control group. The quantity of serum albumin and serum total proteins reduced significantly in treated rabbits on day 10 compared to the control group.

### 3.4. Microscopic Observations

At histological observations, the liver section of rabbits exposed to CIONPs exhibited cytoplasmic vacuolation, disorganized hepatic cord, edema, and necrosis of hepatic nuclei on day 10 of the experiment.

Various microscopic changes such as necrosis and increased urinary spaces, congestion, epithelial detachment from the tubular basement membrane, and necrosis of renal tubules and deposition of casts in the lumen of renal tubules in kidneys of exposed rabbits were examined on day 10 of the research trial. The different sections of heart of exposed rabbits on day 10 of the experimental study indicated different histopathological alterations such as wavy myocardial fibers, necrosis of cardiac myocytes, edema, disorganization of cardiac muscles, and congestions. At the microscopic level, the spleen of exposed rabbits revealed various histological ailments such as the red pulp containing scattered lymphocytes, deposition of ceroid, depletion of white pulp, and increased red pulp. The brain of exposed rabbits exhibited atrophy and degeneration of Purkinje neurons in cerebellum, congestion, pyknosis of neurons, and microgliosis at day 10 of the experimental study (Figure 8 and Figure 9).

## 4. Discussion

Among the several ferrites, CIONPs have undergone some of the most extensive biological investigations. Additionally, the European Medicines Agency (EMA) and the U.S. Food and Drug Administrations (FDA) have approved their usage as MRI contrasts [52]. According to the findings of these investigations, CIONPs may be among the greatest options for potential use in biological areas.

These MNPs used in biological applications should be highly paramagnetic at room temperature [53,54]. Due to their high saturation magnetization, CIONPs hold promise as a base for drug delivery systems and other medicinal uses such as MRI contrast agents. Additionally, they are promising prospects for use in biological applications due to their strong magneto-crystalline anisotropy [55,56]. Additionally, it has been shown that nanoparticles are widely employed in the biomedical sciences because to their low toxicity and outstanding physiochemical properties, including biocompatibility, stability in aqueous solutions, and superparamagnetism [57,58].

Cobalt iron oxide (CoFe_2_O_4_) nanoparticles (CIONPs) with an average size of 13.2 nm were created for this investigation. In our analysis, the CIONPs’ diffraction peaks were sharper and narrower, suggesting that the crystallinity and crystallite size had increased. Using magnetic field strengths ranging from 12,000 Oe to 12,000 Oe, the magnetic properties of CIONPs were examined at room temperature. The Ms value was around 50.1 emu g^−1^, which fits its hyper paramagnetic nature well [59,60,61]. Excellent saturation magnetization enhanced with the concentration of cobalt in iron oxide, suggesting that these nanoparticles may play a notable role in the field of diagnostic methods to see the pathology of different organs in the body. However, the Ms value can be increased after mechanically milling or by doping metals with other ferrites [61,62,63].

In vivo, MRI results exhibited a strong contrast of the liver and stomach, indicating that CIONPs may play a significant role, even at low-field MRI units, in diagnostic modality to identify diseases of organs in the body. Contrast agent choice is based on the relaxivity ratio r_2_/r_1_, which can be enhanced depending on the size, charge concentration, and field strength [24,64,65,66]. T_1_ relaxation seems to be a little faster in the liver than in the spleen based on relaxivity, and T_2_ relaxation also appears to be slightly faster in the liver than in the spleen [39]. Earlier research, however, has found different relaxation rates in the liver and spleens of rats and rabbits. Consequently, the proposed CIONPs could be used as negative contrast agents in diagnostic imaging to reveal organ pathology even with limited-field MRI equipment. According to earlier studies [67,68,69], CIONPs with many of these properties are good for in vivo biomedical fields such as the MRI contrast agent and relaxivity.

The hematological analysis in this study revealed lower values of red blood cells, hemoglobin quantity, and hematocrit percent, with increased levels of total leukocytic count and neutrophils in treated rabbits. The lower blood values could be due to the adverse effects of cobalt iron oxide nanoparticles on bone marrow, whereas the increased levels of white blood cells and neutrophils might be due to injurious stimuli and the generation of free radicals [70,71]. The increased biomarkers of renal function tests, liver function tests, and cardiac biomarkers in our study may be due to hepatic, renal, and cardiac damage [72]. Previously, an increased induction of oxidative stress, the generation of free radicals, and hepatic lipid peroxidation have been recorded due to magnetic nanoparticles [19,72]. In contrast to our results, no significant alterations in serum alanine aminotransferase difference and aspartate aminotransferase were measured due to the nanoparticles. The exposure to iron salts and Fe_3_O_4_ NPs significantly increases lymphocytes, blood cell counts, and goblet cells of the intestine of fish [73]. A significantly increased quantity of oxidative stress and depletion of antioxidant biomarkers due to the exposure to iron oxide nanoparticles was recorded in our study. Previously, a reduction in the activity of catalase and increase in the amount of acid phosphatase, glutathione-s-transferase, and oxidative stress biomarkers at low doses of cobalt ferrite nanoparticles were estimated in fish [73,74]. The changes in biomarkers of oxidative stress and antioxidant parameters in the present study may be due to abnormalities in the mitochondrial respiration in the exposed tissues of rabbits. Studies have recorded significantly increased lipid peroxidation and ROS in the muscles of rats and fish [6]. The significantly increased frequency of DNA damage in hepatocyte, cardiac myocytes, and isolated cells of the kidneys in iron-oxide-exposed rabbits at day 10 could be because of the activation of nuclear factor kappa (NF-kB), resulting in increased oxidative stress and the fragmentation of DNA material [75,76,77].

Different histopathological abnormalities such as degeneration of the hepatocyte, congestion, edema, and necrosis of the hepatocyte in our study have also been observed in a study in rats due to iron oxide nanoparticle treatments in liver tissues [78]. These ailments in treated animals might be due to the activation of different prototypical AhR-regulated genes (Cyp1a1, Nqo1 mRNA, and Cyp1b1) in hepatocytes leading to nuclear localization as a result of the rapid induction of oxidative stress [79,80]. Furthermore, these abnormal necrotic alterations in the liver of nanoparticle-treated rabbits could also be due to the rapid generation of free radicals resulting in the release of different apoptotic signals from the mitochondrial membranes causing necrosis of the liver parenchyma [81]. In the published literature, little information is available regarding the deleterious effects of Fe_3_O_4_ nanoparticles in rabbits [19]. However, different previous studies have also observed different pathological changes in the liver of rats, rabbits, and other animals due to nanoparticles [19,82,83]. Previously, various histological abnormalities in liver tissues, such as an increase in sinusoid space, lipidosis, and necrosis at sub-lethal concentrations of oleat-coated iron oxide nanoparticles (OC-Fe_3_O_4_ NPs), were observed in rats [84]. The magnetic nanoparticles (iron oxide) cause little effects on the cell viability, apoptosis, and oxidative stress. At microscopic levels, various microscopic abnormalities observed in our study in the kidneys of nanoparticle-treated rabbits can be related to the acetylation of histone and DNA methylation in association with the increased production of IL-8 and abnormalities in its promoter.

Moreover, previously, no pathological alterations in the kidneys of treated mice due to magnetic iron oxide nanoparticles have been observed [39]. These alterations could also be related to the modulation of CpG resulting in dysregulation and cell migratory competences favoring renal toxicity. Different histopathological alterations such as tubular epithelial cell degeneration, congestion, necrosis, and hemorrhages in the kidneys of various animals exposed to nanoparticles have also been observed [85,86,87]. In the present study, the depletion of splenic cells and degeneration of white and red pulp of the spleen of exposed rabbits were observed under a light microscope. Previously, no histopathological alterations in the spleen of mice exposed to magnetic nanoparticles were observed [39]. The histological changes in the spleen of rabbits might be due to the induction of oxidative stress by iron oxide nanoparticles. The induction of oxidative stress and lipid peroxidation due to nanoparticles has been determined in mice [39]. The histopathological abnormalities observed in our study in different visceral tissues could be related to the over-release of various proteins (tumor necrosis factor receptor-2 and caspase 3 protein). Furthermore, microscopic abnormalities in our study may also be related to an increased production of connective tissue growth factor and collagen fibers proteins after day 10 of exposure to nanoparticles. In our study, different histological changes in the brain, such as the necrosis of neurons and microgliosis, have not been observed due to magnetic oxide nanoparticles. However, studies have shown that nanoparticles do not cross the blood–brain barrier [39]. It has been further stated that 100% of large molecule neurotherapeutics and 98% of small molecule drugs cannot pass the blood–brain barrier through passive diffusion due to the small tight junction gaps [31,88,89,90]. The microscopic abnormities in brain tissues in our study can also be due to the induction of oxidative stress in terms of the rapid production of free radicals (hydrogen peroxide and reactive oxygen species) due to iron oxide nanoparticles. Various microscopic alterations in the heart of iron-oxide-treated rabbits in this study might be related to release of inflammatory mediators and generation of reactive oxygen species resulting in an increased induction of oxidative stress [39].

## 5. Conclusions

In this study, CoFe_2_O_4_ nanoparticles with a size of about 13.2 nm were synthesized by a co-precipitation method. XRD characterization showed excellent confirmation of crystallinity and its phase identification. Magnetic behavior was studied by drawing Ms versus H, showing a high saturation magnetization value of 50.1 emug^−1^. The MRI study concluded that CIONPs increased the signal strength of T_2_ contrast agents. Cobalt iron oxide contrast agents may be used in clinical settings to identify a variety of disorders. The biological evaluation of nanoparticles indicated that cobalt-coated iron oxide nanoparticles induce deleterious effects on the hematology and serum biochemistry, as well as microscopic changes in various visceral tissues of rabbits. This study concludes that cobalt iron oxide (CoFe_2_O_4_) nanoparticles have induced toxicity in rabbits, and these can be used as biomarkers in medical diagnostics.

## Figures and Tables

**Figure 1 vetsci-10-00514-f001:**
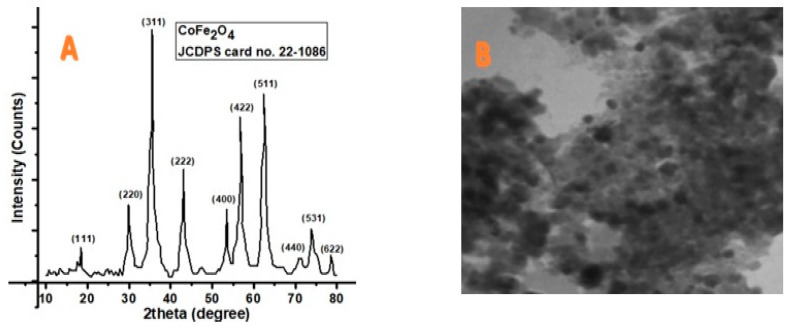
(**A**) XRD result showing phase identification patterns of CoFe_2_O_4_; (**B**) TEM images of CoFe_2_O_4_.

**Figure 2 vetsci-10-00514-f002:**
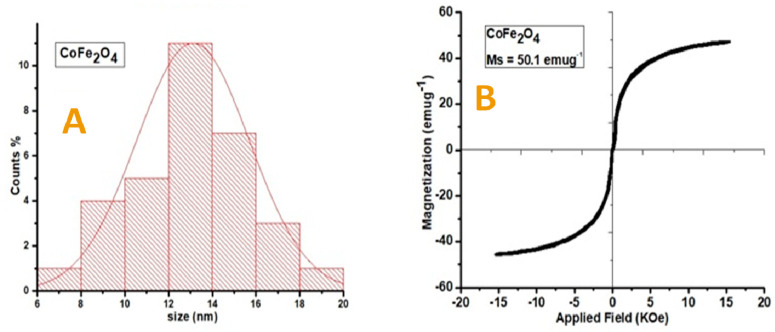
(**A**) Histogram photograph presenting average particle size of CoFe_2_O_4_; (**B**) VSM Plot showing magnetic behavior of CoFe_2_O_4_.

**Figure 3 vetsci-10-00514-f003:**
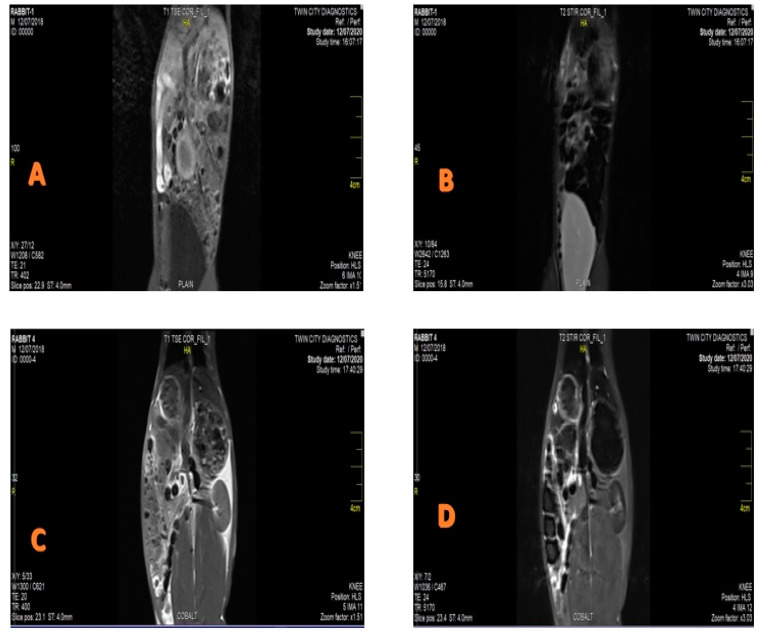
MRI Images of spleen and liver of the (**A**,**B**) control/untreated rabbits presenting no contrast; (**C**,**D**) treated rabbits presenting contrast after exposure of CoFe_2_O_4_ nanoparticles.

**Figure 4 vetsci-10-00514-f004:**
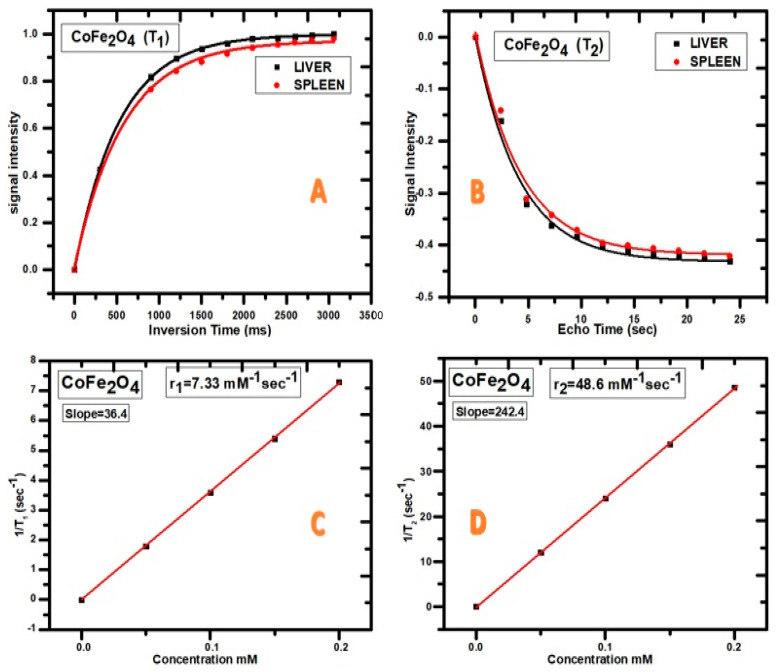
(**A**) Signal intensity for T_1_ weighted, (**B**) signal Intensity for T_2_ weighted, (**C**) relaxivity r_1_, (**D**) relaxivity r_2_.

**Figure 5 vetsci-10-00514-f005:**
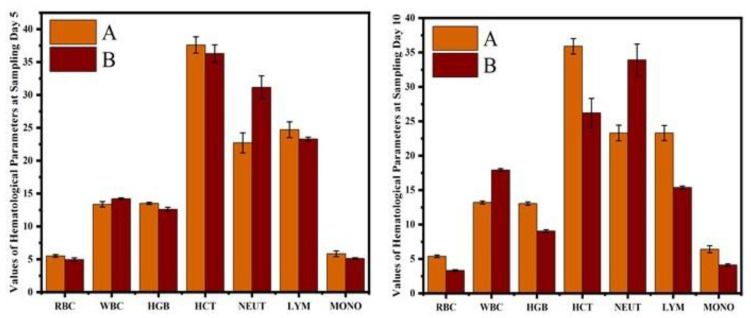
Comparison of hematological profile of normal and cobalt iron oxide (CoFe_2_O_4_) nanoparticles (CIONPs)-treated rabbits at (**A**) day 5 and (**B**) day 10.

**Figure 6 vetsci-10-00514-f006:**
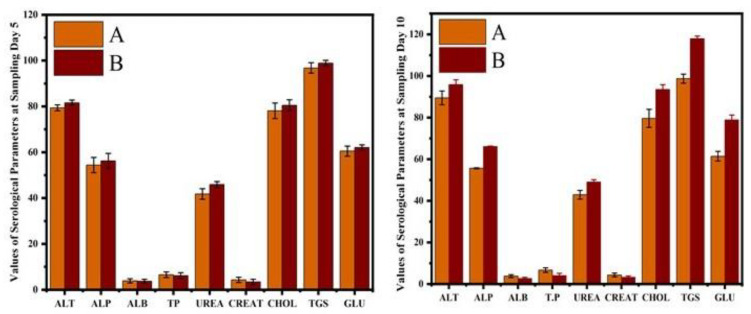
Comparison of serum biochemistry profile of normal and cobalt iron oxide (CoFe_2_O_4_) nanoparticles (CIONPs)-treated rabbits at (**A**) day 5 and (**B**) day 10.

**Figure 7 vetsci-10-00514-f007:**
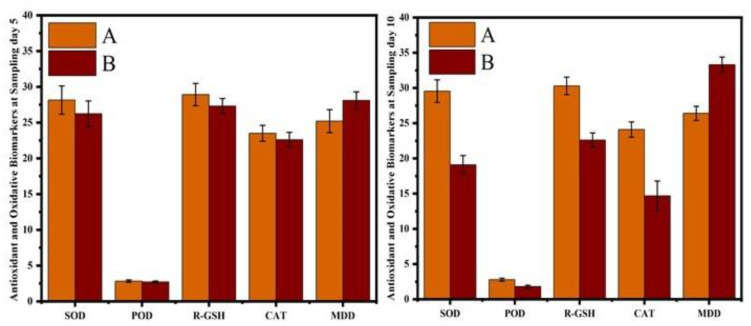
Antioxidant enzymes and oxidative stress biomarkers of normal and cobalt iron oxide (CoFe_2_O_4_) nanoparticles (CIONPs)-treated rabbits at (**A**) day 5 and (**B**) day 10.

**Figure 8 vetsci-10-00514-f008:**
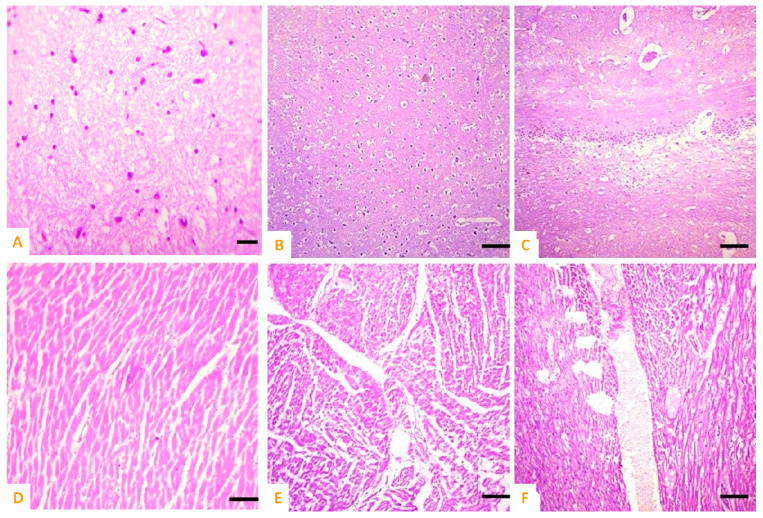
Photomicrograph of (**A**) normal brain of control; (**B**,**C**) cobalt iron oxide (CoFe_2_O_4_) nanoparticles (CIONPs)-treated rabbits exhibiting necrosis, atrophy, and degeneration of neurons in cerebellum, congestion, and microgliosis in brain sections; (**D**) normal heat of control; (**E**,**F**) disorganization of myocardial fibers, necrosis of cardiac myocytes, edema, disorganization of cardiac muscles and congestions.

**Figure 9 vetsci-10-00514-f009:**
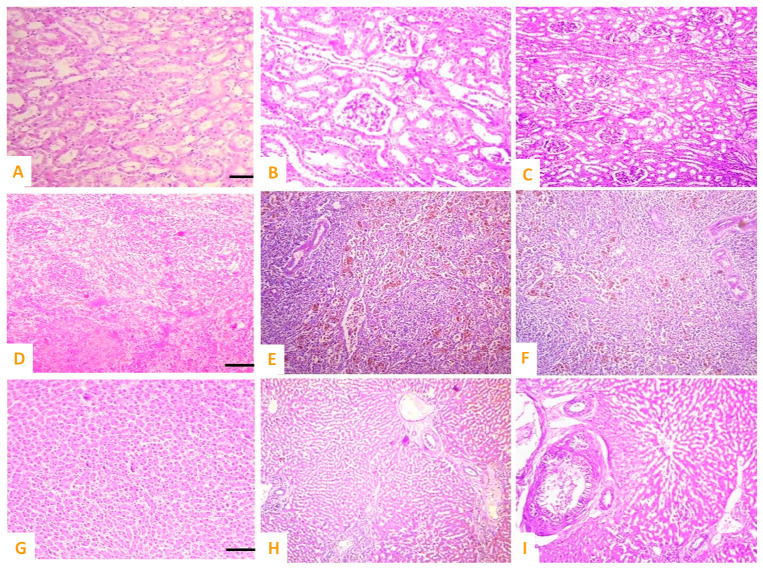
Photomicrograph of (**A**) normal kidneys of control group, cobalt iron oxide (CoFe_2_O_4_) nanoparticles (CIONPs)-treated rabbits exhibiting (**B**,**C**) necrosis and increased urinary spaces, epithelial detachment from the tubular basement membrane, necrosis of renal tubules and deposition of casts in the lumen of renal tubules in kidneys; (**D**) normal spleen of control; (**E**,**F**) red pulp showing scattered lymphocytes, deposition of ceroid, depletion of white pulp and increased red pulp, and (**G**) normal hepatocytes of control group; (**H**,**I**) cytoplasmic vacuolation, disorganized hepatic cord, edema, and necrosis of hepatic nuclei.

## Data Availability

The data presented in this article are available in the article and Appendix A.

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
