# Peer review of "Cobalt Iron Oxide (CoFe2O4) Nanoparticles Induced Toxicity in Rabbits"

_vetsci, 2023, doi:10.3390/vetsci10080514_

Round 1

Reviewer 1 Report

In this manuscript, the authors conducted a comprehensive assessment of the characteristics and in vivo toxicity of Cobalt Iron Oxide (CoFe2O4) Nanoparticles. It’s a very interesting and necessary study for the potential clinical application of these nanoparticles. As we all know, medical device regulatory organizations such FDA would require Good Laboratory Practice (GLP) preclinical studies being performed for medical devices prior to clinical trials. More papers like this one will and should be published. To make the manuscript contents clearer to the readers, I have the following questions and comments for the consideration of authors:

1.       For the preclinical experiment design, can the authors please explain the following:

a.       Why was rabbit selected as the test model?

b.       How relevant is the rabbit model to human?

c.       How were the assessment time points determined (5 days and 10 days)?

2.       In figure 1, the TEM images clearly showed aggregation of the nanoparticles. This is common for non-surface functionalised nanoparticles. Please discuss the impact of particle aggregation on their toxicity.

3.       In the histology figures, please make sure the scale bars are added to the individual images. It would really help the readers if the authors can show the histology images of the control groups at the same regions of interest and clearly label the groups in the figures.

Please check the overall spelling/typos. For example, in section 2.3, TEM should stand for Transmission Electron Microscopy; in line 391, concludes was spelled as ‘concluds’.

Author Response

  1. For the preclinical experiment design, can the authors please explain the following:
  2. Why was rabbit selected as the test model?

Explanation: The physiology of rabbits closely resembles humans as reported in numerous studies.

  1. How relevant is the rabbit model to human?

Explanation: The physiology of rabbits closely resembles humans.

  1. How were the assessment time points determined (5 days and 10 days)?

Explanation: For biological evaluation, the sampling was conducted at day 5 and 10 of post therapy.

  1. In figure 1, the TEM images clearly showed aggregation of the nanoparticles. This is common for non-surface functionalised nanoparticles. Please discuss the impact of particle aggregation on their toxicity.

Explanation: CIONPs used in our study is an example of non-surface functionalized behaviour, which occurs when a surface is not modified in any way to acquire physical, chemical, or biological properties different from those it already possessed. Such materials do not have conjugation or reactive sites that facilitate the formation of covalent bonds with other molecules. Therefore these nanoparticles were used for evaluation of toxicity on multiple tissues of rabbits.

  1. In the histology figures, please make sure the scale bars are added to the individual images. It would really help the readers if the authors can show the histology images of the control groups at the same regions of interest and clearly label the groups in the figures.

Explanation: Now in the histology figures the scale bars are added.

Comments on the Quality of English Language

Please check the overall spelling/typos. For example, in section 2.3, TEM should stand for Transmission Electron Microscopy; in line 391, concludes was spelled as ‘concluds’.

Explanation: Revised the overall spelling/typos. In section 2.3, TEM stand for Transmission Electron Microscopy; in line 391, concludes spelled has been corrected as ‘concludes’.

Reviewer 2 Report

This paper deals with the synthesis of magnetic nanoparticles via the condensation of cobalt iron oxide (CoFe2O4) and its toxic effects on animals (rabbits). The study of this paper concludes that cobalt iron oxide nanoparticles (CoFe2O4) have induced toxicity.

1- A curve at low temperatures would be interesting to recheck whether nanoparticles are superparamagnetic at room temperature.

2-Figure 1 could be divided and form two figures. In this way we could verify in more detail the information.

3-How many TEM images and how many particles were used to form the size distribution figure?

4-Figure 2 It is not possible to verify the information contained in the figure. It seems to be out of focus.

5-All figures must contain the same format in the information of Figure (a), Figure (b) ... (d).

6-The speeches are long but without illustrations. I believe that the best thing would be to hold these discussions right after presenting the results. This would make it easier to visualize the explanations.

7-Revision in English is required.

8-The topic is interesting but the work needs more results to conclude the results on the subject on induced toxicity this part of the work was not clear.

Revision in English is required.

Author Response

1- A curve at low temperatures would be interesting to recheck whether nanoparticles are superparamagnetic at room temperature.

Explanation: The magnetic behavior of the synthesized nanoparticles was confirmed by VSM. The results indicated that CIONPs have excellent saturation magnetization value of about 50.1 emug-1 and coercivity is almost zero. The magnetization hysteresis reveals that the CIONPs had super paramagnetic nature.

2-Figure 1 could be divided and form two figures. In this way we could verify in more detail the information.

Explanation: Followed these instructions in this revised manuscript.

3-How many TEM images and how many particles were used to form the size distribution figure?

Explanation: Now we have included this information in this revised manuscript.

4-Figure 2 It is not possible to verify the information contained in the figure. It seems to be out of focus.

Explanation: Please accept this figure 2, please accept it as it is. Or suggest that we may remove that Figure 2?

5-All figures must contain the same format in the information of Figure (a), Figure (b) ... (d).

Explanation: Followed these instructions in this revised manuscript.

6-The speeches are long but without illustrations. I believe that the best thing would be to hold these discussions right after presenting the results. This would make it easier to visualize the explanations.

Explanation: Followed these instructions in this revised manuscript.

7-Revision in English is required.

Explanation: Now this paper has been revised with the help of native English expert.

8-The topic is interesting but the work needs more results to conclude the results on the subject on induced toxicity this part of the work was not clear.

Explanation: Now this paper is revised according the suggested instructions.

Comments on the Quality of English Language

Revision in English is required.

Explanation: Revised the English.

Reviewer 3 Report

Dear Editor,

The manuscript entitled “Cobalt Iron Oxide (CoFe2O4) Nanoparticles Induced Toxicity in Rabbits” by Muhammad Shahid Khan et al. presents a study of cobalt iron oxide (CoFe2O4) nanoparticles (CIONPs) synthesis and evaluation of their toxic effect in rabbits. The authors have fully characterized the synthesized CIONPS and have concluded that the cobalt iron oxide nanoparticles have induced toxicity in rabbits.

Τhe manuscripts’ objects are interesting, it written in a comprehensive way and the findings are interesting, however the authors statements that the CIONPs could be used as biomarkers in medical diagnostics seems unjustified based on the rabbits toxicity results.  Therefore, the manuscript could be accepted for publication after major revisions. My specific comments are:

1.      Most of the references used for the introduction section are older than 2016. The authors should update their information based in more recent references to provide the state of the art in the field of study.

2.      Please check reference 11; the authors’ statement doesn’t look in the right context.

3.      The paragraph in page 2, lines 70-80, should be removed; provides many details on MRI parameters without explaining why it is needed here.

4.      All material and methods need to be re-written to provide more details of the used methods and the used instrumentation. Providing a reference cannot substitute a method description.

5.      In section 2.6, how the CIONPs were given to the rabbits? What was given to control groups instead of CIONPs?

6.      Please add a statistical analysis for the obtained results, to find significant differences in the studied groups.

7.      Please add size bars in figures, where appropriate.

8.      Instead of tables 1 -3, the authors should provide graphs and move the tables as supplementary materials.

9.      The discussion section needs to be re-written: it contains much information but it is difficult to understand the authors’ points and connection to the study findings. The CIONPs advantages should be stated more clearly. Also, I don’t understand how the authors discuss extensively for DNA damage. I couldn’t understand the assay results they were based for such an extensive discussion on DNA damage.

10.  The authors should provide references with all authors, not just the first author.

Author Response

Most of the references used for the introduction section are older than 2016. The authors should update their information based in more recent references to provide the state of the art in the field of study.

Explanation: Now this manuscript has been updated and the recent references are added.

  1. Please check reference 11; the authors’ statement doesn’t look in the right context.

Explanation: Followed these instructions in this revised manuscript.

  1. The paragraph in page 2, lines 70-80, should be removed; provides many details on MRI parameters without explaining why it is needed here.

Explanation: Followed these instructions in this revised manuscript.

  1. All material and methods need to be re-written to provide more details of the used methods and the used instrumentation. Providing a reference cannot substitute a method description.

Explanation: Followed these instructions in this revised manuscript.

  1. In section 2.6, how the CIONPs were given to the rabbits? What was given to control groups instead of CIONPs?

Explanation: These particles were intravenously injected while normal saline was given to rabbits.

  1. Please add a statistical analysis for the obtained results, to find significant differences in the studied groups.

Explanation: Followed these instructions in this revised manuscript.

  1. Please add size bars in figures, where appropriate.

Explanation: Followed these instructions in this revised manuscript.

  1. Instead of tables 1 -3, the authors should provide graphs and move the tables as supplementary materials.

Explanation: Followed these instructions in this revised manuscript.

  1. The discussion section needs to be re-written: it contains much information but it is difficult to understand the authors’ points and connection to the study findings. The CIONPs advantages should be stated more clearly. Also, I don’t understand how the authors discuss extensively for DNA damage. I couldn’t understand the assay results they were based for such an extensive discussion on DNA damage.

Explanation: The discussion section is re-written: Now will be easy to understand our points and connection to the study findings. The CIONPs advantages are rewritten more clearly. Discussion on DNA damage is revised.

  1. The authors should provide references with all authors, not just the first author.

Explanation:  The references are according to the format of journal in revised submission.

Round 2

Reviewer 3 Report

Dear editor, the authors have incorporated most of my comments on their revised manuscript therefore I suggest publication in its present form